

# Multimodal sensorimotor assessment of hand and forearm asymmetries: a reliability and correlational study

Pablo Bellosta-López[1],[*], Julia Blasco-Abadía[1],[*], Lars L. Andersen[2],[3], Jonas Vinstrup[2], Sebastian V. Skovlund[2],[4] and Víctor Doménech-García[1]

[1] Faculty of Health Sciences, Universidad San Jorge, Villanueva de Gállego, Zaragoza, Spain
[2] National Research Centre for the Working Environment, Copenhagen, Denmark
[3] Department of Health Science and Technology, Aalborg University, Aalborg, Denmark
[4] Department of Sports Science and Clinical Biomechanics, University of Southern Denmark, Odense, Denmark
[*] These authors contributed equally to this work.

Corresponding authors
Pablo Bellosta-López,
pbellosta@usj.es
Víctor Doménech-García,
vdomenech@usj.es

## ABSTRACT

**Background:** Effective rehabilitation of upper limb musculoskeletal disorders requires multimodal assessment to guide clinicians' decision-making. Furthermore, a comprehensive assessment must include reliable tests. Nevertheless, the interrelationship among various upper limb tests remains unclear. This study aimed to evaluate the reliability of easily applicable upper extremity assessments, including absolute values and asymmetries of muscle mechanical properties, pressure pain threshold, active range of motion, maximal isometric strength, and manual dexterity. A secondary aim was to explore correlations between different assessment procedures to determine their interrelationship.

**Methods:** Thirty healthy subjects participated in two experimental sessions with 1 week between sessions. Measurements involved using a digital myotonometer, algometer, inclinometer, dynamometer, and the Nine-Hole Peg test. Intraclass correlation coefficients, standard error of the mean, and minimum detectable change were calculated as reliability indicators. Pearson's correlation was used to assess the interrelationship between tests.

**Results:** For the absolute values of the dominant and nondominant sides, reliability was 'good' to 'excellent' for muscle mechanical properties, pressure pain thresholds, active range of motion, maximal isometric strength, and manual dexterity. Similarly, the reliability for asymmetries ranged from 'moderate' to 'excellent' across the same parameters. Faster performance in the second session was consistently found for the Nine-Hole Peg test. No systematic inter-session errors were identified for the values of the asymmetries. No significant correlations were found between tests, indicating test independence.

**Conclusion:** These findings indicate that the sensorimotor battery of tests is reliable, while monitoring asymmetry changes may offer a more conservative approach to effectively tracking recovery of upper extremity injuries.

## INTRODUCTION

Musculoskeletal disorders (MSDs) in the upper extremities have a high prevalence globally, especially among workers (*Govaerts et al., 2021*) and athletes (*Lehman et al., 2020*). Because humans are so dependent on their hands, MSDs in the upper extremities often cause greater disability and reduced health-related quality of life compared to other body regions (*de Putter et al., 2014*). These conditions can prevent normal functioning in daily activities, especially when the dominant wrist and hand are affected (*Jayakumar et al., 2018*). Additionally, MSDs in the upper extremities are robustly associated with high costs related to healthcare services and lost productivity (*de Putter et al., 2012*).

Professionals involved in the rehabilitation process face challenges in determining when an athlete or a worker with upper limb MSDs is ready to return to training (*Oak et al., 2022*) or work (*Hosseininejad et al., 2023*), respectively. Physical demands on the upper limb often require preserved functional properties such as strength, movement amplitude, and dexterity, as well as normal pain sensitivity. For example, it is well known that the sensation of tissue tightness or pain can limit sports or work performance (*Barr, Barbe & Clark, 2004*). Therefore, a comprehensive and easily applicable assessment of objective measures of pain sensitivity and physical performance holds great potential in improving current rehabilitation practices. However, tests composing multimodal assessment protocols are usually conducted in different study samples, impeding comparison between tests' reliability (*Llanos et al., 2021*; *Schrama et al., 2014*). Furthermore, one of the challenges in the repeated assessment of physical performance tests is the presence of learning effects and/or skill improvement in the technical execution of the tasks (*Tsigilis & Theodosiou, 2008*), which likely varies between the dominant and non-dominant sides (*Stöckel & Weigelt, 2012*). For example, the non-dominant side may have a greater learning potential, which can skew asymmetry indices with repeated assessments over time. This effect is particularly problematic in the context of rehabilitation, as it can lead to false indications of improvement that may not genuinely exist or be insignificant. To address this issue, rehabilitation professionals commonly perform bilateral assessments of both limbs during various musculoskeletal examinations (*Walker, Hall & Hurst, 1990*). This approach allows rehabilitation professionals to determine the presence of a relevant imbalance or discrepancy between the two sides, providing valuable insights into the subject's condition (*Jones, 2019*), especially when beyond the normal asymmetries within a reasonable range of 10% (*Evershed, Burkett & Mellifont, 2014*). However, despite the widespread use of asymmetry assessment, the reliability of such an approach has not been well explored. In this sense, assuring the reliability of any assessment procedure is essential for its implementation in clinical and research practices (*Kimberlin & Winterstein, 2008*).

This study aimed to evaluate the reliability of a range of easy-to-implement procedures assessing muscle mechanical properties, manual dexterity, pressure pain thresholds, active range of motion, and maximal isometric strength of the upper limbs in terms of absolute values and asymmetries between both sides. Furthermore, the correlations between the different assessment procedures were explored as a secondary aim to determine the interrelationship between tests.

## MATERIALS AND METHODS

### Design

This reliability study with a repeated-measures design was conducted at the biomechanics lab of the university campus of San Jorge University, Spain, in accordance with the Helsinki Declaration and approved by the regional ethics committee "Comité de Ética de la Investigación de la Comunidad Autónoma de Aragón" (C.P.-C.I. PI18/385). All participants provided written informed consent before entering the study. The present study has been reported following the Guidelines for Reporting Reliability and Agreement Studies (GRRAS) (*Kottner et al., 2011*). The methodology and procedures used in the present study were adapted from previous studies on multimodal sensorimotor assessment of the lower extremities (*Doménech-García et al., 2023*; *Bellosta-López et al., 2023*; *Bellosta-López et al., 2024*).

### Participants

Participants were recruited from the local community through advertisements at the university campus and on social media. The inclusion criteria were: (i) age 18 to 50 years and (ii) absence of pain and functional limitations due to past or current injuries or pathologies in the upper extremities. Exclusion criteria were: (i) a history of pain at any part of the body in the previous 3 months; (ii) presence of chronic pain (*e.g.*, chronic low back pain, fibromyalgia, migraine); (iii) a history of serious injury to the upper extremities (*e.g.*, fracture, surgery); (iv) chronic use of medication; and (v) diagnosis of serious diseases. The rationale for recruiting healthy participants was to ensure consistency in participant status across the first and second assessments for test-retest reliability. Participants were instructed to avoid engaging in any physical activity or exercise that was unaccustomed or of high intensity throughout the study period.

### Sample size

A total of 30 participants were needed to reach a power of 90% with an alfa error of 0.05, expecting a 'good' to 'excellent' intraclass correlation coefficient (ICC) values between 0.70 and 0.90 for a single measurement two-way mixed model (*Walter, Eliasziw & Donner, 1998*).

### Procedure

Participants attended two test sessions separated by a 7-day interval. The assessments took place at the same time of day to ensure consistent testing conditions (*Bellosta-López et al., 2023*). Following a standardized protocol, the same assessor, who was trained in the assessment protocol, conducted all the test sessions. An external supervisor provided oversight to ensure methodological quality and consistency. All sessions were conducted in a controlled environment, in a quiet room with consistent ambient light, and temperature and humidity controlled. After verifying the inclusion criteria and obtaining informed consent, socio-demographic data and anthropometric characteristics were collected. Anthropometric measurements were performed according to the International Society for

Advancement of Kinanthropometry (*International Society for Advancement of Kinanthropometry I, 2001*) and included forearm and wrist circumferences, forearm length, humerus bi-epicondylar and wrist bi-styloid diameters, and forearm skinfold. During the anthropometric assessment, reference points (detailed later in each assessment procedure) were identified by palpation of anatomical landmarks. These points were marked on the skin with semi-permanent ink to facilitate the repeatability of the measurements. The standardized protocol included the following tests, administered in the specified order to minimize the influence of preceding tests on the performance of the subsequent ones: myotonometry, the Nine-Hole Peg Test, algometry, inclinometry, and dynamometry. Each test session lasted approximately 25 min, and the assessor was blinded to findings obtained in previous sessions. The side order (*i.e.*, dominant, non-dominant) was randomized for each participant before the first session and maintained across all assessments.

## Testing positions

The position of the participants was controlled during the execution of all tests (Fig. 1). In Testing Position 1, the participant was seated at the edge of the table with knees bent at 90°, lower back supported by the backrest, shoulders in slight flexion, elbows bent at 90°, palms of the hand supported, and fingers relaxed. The distance between the trunk and the edge of the table was a closed fist, equivalent to approximately 10 cm (Fig. 1A). Testing Position 2 consisted of rotating 90° from 'Testing Position 1' so that the proximal two-thirds of the forearm remained resting on the table, with the distal third and wrist remaining free in the air (Fig. 1B).

## Muscle mechanical properties

Mechanical properties of the muscle were assessed with a hand-held digital myotonometer (MyotonPRO, Müomeetria AS, Estonia), a non-invasive and painless tool that provides objective measurements of the mechanical properties of the muscle *via* the impulses it exerts on the tissue (*Leonard et al., 2003*). The study focused on (i) oscillation frequency (Hz), as an indicator of muscle tone characterizing the resting tension level of the tissue; (ii) dynamic stiffness (N/m), reflecting the tissue resistance to the force deforming the muscle (mechanical impulse); and (iii) logarithmic decrement (arbitrary unit), characterizing the ability of the muscle to return to its initial position after an external force perturbation and corresponding to the elasticity of the tissue (*Aird, Samuel & Stokes, 2012*). Tests were performed with participants positioned in Testing Position 1. Measurements were performed bilaterally on the extensor carpi radialis brevis muscle at its most prominent point following the procedure described by *Riek, Carson & Wright (2000)*. This point was approximately located in the first third of the line connecting the lateral epicondyle with the styloid of the radius. The measurements were taken after requesting a counter-resistance wrist extension to visualize the muscle mass and different partitions, considering the anatomical sequence of the common extensor of the fingers, extensor carpi radialis brevis, extensor carpi radialis longus, and brachioradialis (*Riek, Carson & Wright, 2000*). The assessor ensured there was no tension on the skin, and the probe was

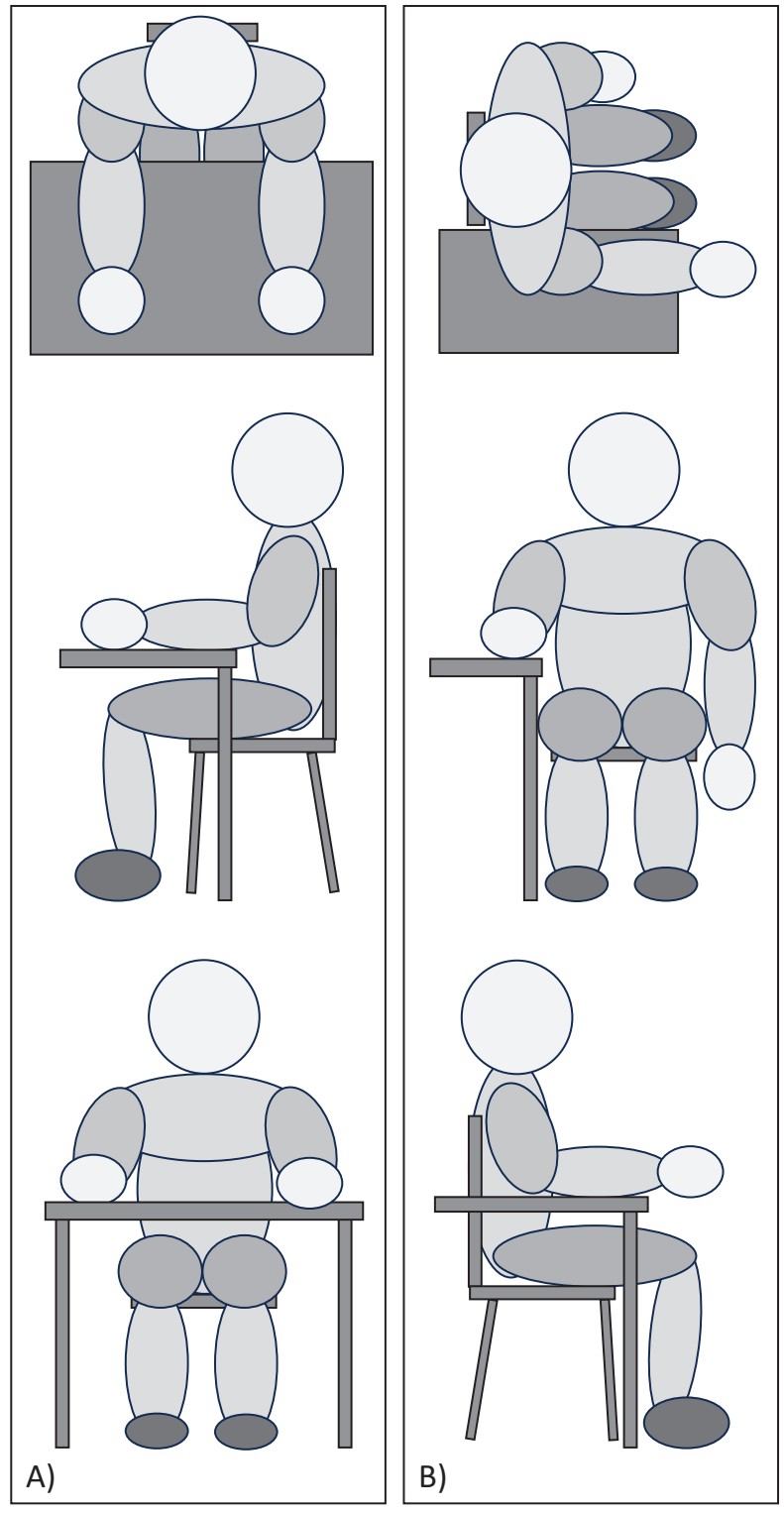

**Figure 1 Picture of (A) Testing Position 1 and (B) Testing Position 2.**

perpendicular to the tissue surface for each measurement. One measurement set of ten consecutive impulses (scan mode) was conducted at each of the marked points with a time interval of 1 s between each impulse. The average value of each series was accepted if the coefficient of variation of the measurement set was less than 3%.

## Manual dexterity

Manual dexterity was assessed using the Nine-Hole Peg test. This test was performed using a plastic commercially available version (Smith & Nephew, Watford, United Kingdom) with nine holes on one side and a hole with nine pegs on the other side. The objective was to insert the pegs into the holes using the hand as quickly as possible (*Oxford Grice et al., 2003*). Participants performed the test in Testing Position 1, keeping the palm of the opposite hand on the table. The position of the box was standardized for each participant, being placed 20 cm from the edge of the table and aligned with the center of the body (*i.e.*, perpendicular to the sternum). After a familiarization trial for each side, three trials were performed for each side. The test completion time was obtained with a stopwatch, and the average value of the three attempts was extracted for reliability analysis. If any of the pegs jumped out of the hole or fell to the ground during the course of the test, that attempt was discarded and a new one was performed.

## Pressure pain thresholds

A digital handheld pressure algometer (Somedic, Hörby, Sweden) mounted with a 1 cm$^2$ probe and connected by a cable to a pushbutton was used to assess pressure pain thresholds. A pre-established familiarization protocol was followed at each session (*Bellosta-López et al., 2023*). The assessor explained to the participants that 'Pressure pain threshold is the pressure reached when you first feel that the pressure becomes painful'. Pressure pain thresholds were assessed bilaterally over two muscles with participants in the Testing Position 1: (i) extensor carpi radialis brevis muscle, at the same point as for myotonometry and (ii) deltoideus muscle, as a control point in the upper extremity at the midline between the acromion and the deltoid tuberosity of the humerus. The pressure was gradually increased at the stimulation site with a ramp of 30 kPa/s. Two measurements of pressure pain thresholds were recorded for each site with a 30-s interval before assessing the same site again. The average value for each site was extracted for reliability analysis.

## Active range of motion

A handheld digital inclinometer (microFET 3®, Hogan Health Industries, Salt Lake City, UT, USA) was used to assess the active range of motion of the wrist (*Kolber & Hanney, 2012*). Participants were asked to actively perform maximal flexion and extension movements from Testing Position 2, stabilizing their forearm with the opposite hand to avoid compensatory movements. The angle between the vertical and the wrist was recorded by placing the inclinometer on the third metacarpal. The inclinometer was calibrated on a horizontal surface before the assessment of each side. Participants had to hold the maximum range of motion for 3 s without shaking for the measurement to be considered valid. During wrist flexion movements, special care was taken to ensure that no

finger flexion occurred. Three consecutive trials were performed interspersed by an interval of 10 s. The average angle value, expressed in degrees, was extracted for further analysis.

## Maximal isometric strength

A handheld digital dynamometer (microFET 3®, Hogan Health Industries, Salt Lake City, UT, USA) mounted with an arc-shaped piece adapted to the metacarpophalangeal region was used to assess the maximal isometric strength of the extensor muscles of the wrist (*Schrama et al., 2014*). Participants were placed in Testing Position 1 with the forearm resting over a wedge with a 30-degree tilt (*Doménech-García et al., 2020*). The wrist was positioned at its edge to allow a slight wrist flexion-extension range before starting to exert force. The forearm was fixed to the table with a strap at the upper third of the forearm. A second strap was adjusted in such a way that, when the dynamometer was placed, the angle of the wrist and the edge allowed a mechanical advantage during the isometric contraction of 5–10 degrees of extension (*Schrama et al., 2014*). A mobile application (Tabata Timer: Interval Timer, Oleksandr Serhiienko, Ukraine) was used to establish an audible and visual preparation countdown of 3 s and a worktime of 5 s to ensure that the muscle contraction had a similar length over repetitions. Participants were also asked to keep the opposite arm relaxed to avoid compensatory movements. Before data collection, a submaximal familiarization trial was performed for each side. Finally, participants performed three repetitions on each side, with an interval of 90 s between assessments of the same side. Visual inspection of the force-time curve was performed to prevent values from distorting the results. The average maximal isometric force value was extracted for further analysis.

## Statistical analysis

Asymmetries were calculated by subtracting the non-dominant value to the dominant value and dividing by the maximum value of one of these two values (*i.e.*, (dominant − non-dominant)/(Max. dominant, non-dominant)). This method to calculate asymmetries is considered one of the most appropriate due to its ability to express the magnitude and direction of asymmetry, thereby overcoming the limitations associated with selecting a reference limb (*Parkinson et al., 2021*).

The Shapiro-Wilk test was used to determine the normal distribution of the sample, and data were presented in terms of mean and standard deviation.

Mixed-model repeated measures analysis of the variance (RM-ANOVA) for the raw values with *day* (assessment 1 and assessment 2) as within factor and *side* (dominant, non-dominant) as between factor were performed to explore the absence of systematic bias (*i.e.*, fixed and proportional bias (*Ludbrook, 2010*)), as well as consistently explore significant between-side differences within tests. Besides, for the asymmetries, paired Student T-tests were performed to explore the presence or absence of systematic bias. When significant, data were presented as mean difference (MD) and 95% confidence intervals (95% CI).

Reliability was evaluated for single measurement absolute agreement based on a two-way mixed model by computing intraclass correlation coefficients ($ICC_{3,1}$). The ICCs were calculated both for the raw values in the non-dominant and the dominant side, as

well as for asymmetries. An ICC above 0.90 was considered as 'excellent', 0.75–0.90 as 'good', 0.50–0.75 as 'moderate', and less than 0.50 as 'poor' reliability (*Koo & Li, 2016*). The standard error of measurement (SEM) was calculated using the following formula: $\text{SEM} = \text{SD}_{\text{pooled}} \times \sqrt{(1\text{-ICC})}$. The SEM represented the expected random variation in scores within a subject when no real change has happened (*Furlan & Sterr, 2018*). The minimum detectable change (MDC) at 95% ($\text{MDC}_{95}$) and 90% ($\text{MDC}_{90}$) was calculated using the formulas: $\text{MDC}_{95} = 1.96 \times \text{SEM} \times \sqrt{2}$; $\text{MDC}_{90} = \text{SEM} \times \sqrt{2} \times 1.64$. The MDC is considered the minimal change needed for being a real change in a sample rather than a random measurement error (*Furlan & Sterr, 2018*). MDC was also calculated at the 90% level due to is considered adequate for measuring a change in ordinary clinical practice (*Donoghue & Stokes, 2009*).

Pearson's correlation was used to assess the interrelationship between the raw values in the non-dominant and the dominant side, as well as for asymmetries in each test procedure. Correlations were considered as 'strong' ($\rho \geq 0.70$), 'moderate' ($0.40 > \rho < 0.69$), 'weak' ($0.10 > \rho < 0.39$), and 'negligible' ($\rho < 0.10$) (*Schober, Boer & Schwarte, 2018*).

Statistical analysis was performed using SPSS v.28 (IBM, Chicago, IL, USA), and a significance level of $P < 0.05$ was accepted. However, due to the multiple comparisons (*i.e.*, nine comparisons per variable), the significant level for the correlation analysis was corrected to $P < 0.006$ (*i.e.*, 0.05 divided by 9).

## RESULTS

Thirty healthy participants (50% female) with an average age of 30 ± 5 years completed the study (Table 1).

### Presence or absence of systematic bias

No *side** *day* interaction was found in the RM-ANOVA for any of the variables ($P > 0.114$), indicating a similar pattern across assessments of the dominant and non-dominant sides. A *day* effect was only found for the Nine Hole Peg test, revealing that the time for completing the task during the assessment-2 was less than during the assessment-1 (MD: 0.39 s; 95% CI [0.19–0.60]; $P < 0.001$). A *side* effect was observed for some variables, indicating that the dominant side performed with less time in the Nine Hole Peg test (MD: 0.92 s; 95% CI [0.68–1.16]; $P < 0.001$), exhibited lower active range of motion for wrist extension (MD: 3.2°; 95% CI [1.7–4.7]; $P < 0.001$) and total range (MD: 2.1°; 95% CI [0.3–3.9]; $P = 0.026$), as well as higher maximal isometric strength (MD: 15.4 N; 95% CI [11.0–19.9]; $P < 0.001$) compared to the non-dominant side (Table 2).

Paired Student T-tests showed no differences across assessments for asymmetries in any of the included variables ($P > 0.123$) (Table 2).

### Test-retest reliability

Table 3 presents $\text{ICC}_{3,1}$, SEM, $\text{MDC}_{90}$, and $\text{MDC}_{95}$ for each variable. For the muscle mechanical properties of the dominant and non-dominant sides, reliability was 'excellent' for the frequency and stiffness parameters (ICC: 0.91–0.96), and 'good' to 'excellent' for the decrement (ICC: 0.86–0.91). For the manual dexterity, reliability was 'good' on both sides

**Table 1 Sociodemographic data, anthropometric characteristics, and physical activity levels of participants.**

| | |
|---|---|
| Age (years) | 30.3 (5.2) |
| Female (n, %) | 15%, 50% |
| Weight (kg) | 71.6 (11.2) |
| Height (cm) | 173.4 (9.8) |
| BMI (kg/m$^2$) | 23.8 (2.7) |
| Forearm circumference D (cm) | 25.7 (2.4) |
| Forearm circumference nD (cm) | 25.4 (2.2) |
| Wrist circumference D (cm) | 15.8 (1) |
| Wrist circumference nD (cm) | 15.8 (1) |
| Forearm length D (cm) | 25.6 (2.5) |
| Forearm length nD (cm) | 25.7 (2.5) |
| Humerus diameter D (cm) | 6.0 (0.6) |
| Humerus diameter nD (cm) | 5.9 (0.6) |
| Wrist diameter D (cm) | 4.9 (0.5) |
| Wrist diameter nD (cm) | 4.8 (0.5) |
| Forearm skinfold D (mm) | 6.7 (2.7) |
| Forearm skinfold nD (mm) | 6.8 (3.1) |

Note:
$N = 30$. Values are represented as mean (standard deviation) unless otherwise specified. D, Dominant; nD, non-dominant; BMI, Body Mass Index.

**Table 2 Clinical characteristics of participants in the different test for the wrist extensor muscles.**

| Measurement test | Side | Assessment 1 | Assessment 2 |
|---|---|---|---|
| Myoton: ECRB frequency (Hz) | Dominant | 15.7 (1.2) | 15.7 (1.1) |
| | Non-dominant | 15.4 (1.4) | 15.3 (1.3) |
| | Asymmetry (%) | 1.6 (5.7) | 2.4 (6.2) |
| Myoton: ECRB stiffness (N/m) | Dominant | 284 (38) | 286 (37) |
| | Non-dominant | 278 (38) | 275 (36) |
| | Asymmetry (%) | 2.0 (10.4) | 3.5 (10.5) |
| Myoton: ECRB decrement (AU) | Dominant | 0.96 (0.11) | 0.98 (0.12) |
| | Non-dominant | 1.00 (0.10) | 1.00 (0.12) |
| | Asymmetry (%) | −3.4 (7.7) | −1.6 (9.3) |
| Nine-Hole Peg test (s) | Dominant | 11.3 (0.8) | 10.9 (0.8)* |
| | Non-dominant | 12.2 (0.9)[#] | 11.8 (1.0)*,[#] |
| | Asymmetry (%) | −7.9 (5.1) | −7.0 (6.4) |
| PPT: ECRB (kPa) | Dominant | 164 (55) | 169 (64) |
| | Non-dominant | 171 (57) | 175 (67) |
| | Asymmetry (%) | −3.1 (19.3) | −2.8 (20.9) |

(Continued)

| Measurement test | Side | Assessment 1 | Assessment 2 |
|---|---|---|---|
| PPT: Deltoideus (kPa) | Dominant | 191 (58) | 184 (55) |
| | Non-dominant | 188 (62) | 196 (79) |
| | Asymmetry (%) | 1.3 (17.4) | −3.6 (16.9) |
| Active range of motion: total (°) | Dominant | 158 (11) | 159 (10) |
| | Non-dominant | 161 (9)[#] | 161 (10)[#] |
| | Asymmetry (%) | −1.5 (3.7) | −1.2 (3.0) |
| Active range of motion: flexion (°) | Dominant | 93 (7) | 93 (7) |
| | Non-dominant | 92 (6) | 92 (7) |
| | Asymmetry (%) | 1.0 (4.8) | 1.1 (3.6) |
| Active range of motion: extension (°) | Dominant | 65 (6) | 66 (6) |
| | Non-dominant | 69 (6)[#] | 68 (6)[#] |
| | Asymmetry (%) | −5.1 (5.8) | −4.1 (6.3) |
| Maximal isometric strength (N) | Dominant | 133 (44) | 133 (42) |
| | Non-dominant | 117 (40)[#] | 119 (40)[#] |
| | Asymmetry (%) | 12.3 (9.8) | 10.9 (9.1) |

**Notes:**
$N$ = 30. Values are represented as mean (standard deviation). ECRB, extensor carpi radialis brevis; PPT, Pressure Pain Threshold.
[*] Significant differences compared to the values in the Assessment 1 with $P < 0.05$.
[#] Significant differences compared to the values in the dominant side with $P < 0.05$.

**Table 3 Reliability indicators of myotonometry, manual dexterity, pressure pain thresholds, active range of motion, and maximal isometric strength for the wrist extensor muscles.**

| Measurement test | Side | ICC (95% CI) | SEM AV (RV) | MDC$_{90}$ AV (RV) | MDC$_{95}$ AV (RV) |
|---|---|---|---|---|---|
| Myoton: ECRB frequency (Hz) | Dominant | 0.96 [0.91–0.98] | 0.2 (1.6) | 0.6 (3.7) | 0.7 (4.4) |
| | Non-dominant | 0.93 [0.85–0.97] | 0.4 (2.3) | 0.8 (5.4) | 1.0 (6.5) |
| | Asymmetry (%) | 0.78 [0.54–0.89] | 2.8 | 6.5 | 7.7 |
| Myoton: ECRB stiffness (N/m) | Dominant | 0.94 [0.87–0.97] | 10 (3.3) | 22 (7.7) | 26 (9.3) |
| | Non-dominant | 0.91 [0.81–0.96] | 11 (4.1) | 26 (9.4) | 31 (11.2) |
| | Asymmetry (%) | 0.72 [0.41–0.87] | 5.5 | 12.8 | 15.4 |
| Myoton: ECRB decrement (AU) | Dominant | 0.91 [0.80–0.96] | 0.04 (3.7) | 0.08 (8.5) | 0.10 (10.1) |
| | Non-dominant | 0.86 [0.70–0.93] | 0.04 (4.2) | 0.10 (9.6) | 0.12 (11.5) |
| | Asymmetry (%) | 0.77 [0.52–0.89] | 4.1 | 9.4 | 11.3 |
| Nine-Hole Peg test (s) | Dominant | 0.82 [0.53–0.92] | 0.3 (3.0) | 0.8 (7.0) | 0.9 (8.3) |
| | Non-dominant | 0.79 [0.46–0.91] | 0.4 (3.6) | 1.0 (8.4) | 1.2 (10.1) |
| | Asymmetry (%) | 0.72 [0.40–0.86] | 3.0 | 7.1 | 8.4 |
| PPT: ECRB (kPa) | Dominant | 0.90 [0.67–0.94] | 19 (11.4) | 44 (26.4) | 53 (31.6) |
| | Non-dominant | 0.93 [0.84–0.96] | 17 (9.8) | 39 (22.7) | 47 (27.1) |
| | Asymmetry (%) | 0.58 [0.30–0.80] | 13.0 | 30.2 | 36.1 |
| PPT: Deltoideus (kPa) | Dominant | 0.84 [0.66–0.92] | 23 (12.3) | 53 (28.4) | 64 (34) |
| | Non-dominant | 0.89 [0.76–0.95] | 24 (12.3) | 55 (28.6) | 66 (34.2) |
| | Asymmetry (%) | 0.53 [0.33–0.77] | 11.8 | 27.4 | 32.7 |

| Table 3 (continued) | | | | | |
|---|---|---|---|---|---|
| Measurement test | Side | ICC (95% CI) | SEM AV (RV) | MDC$_{90}$ AV (RV) | MDC$_{95}$ AV (RV) |
| Active range of motion: total (°) | Dominant | 0.96 [0.92–0.98] | 1.9 (1.2) | 4.5 (2.9) | 5.4 (3.4) |
| | Non-dominant | 0.97 [0.94–0.99] | 1.6 (1.0) | 3.8 (2.4) | 4.5 (2.8) |
| | Asymmetry (%) | 0.73 [0.43–0.87] | 1.7 | 4.0 | 4.8 |
| Active range of motion: flexion (°) | Dominant | 0.95 [0.90–0.98] | 1.6 (1.7) | 3.6 (3.9) | 4.3 (4.7) |
| | Non-dominant | 0.96 [0.91–0.98] | 1.4 (1.5) | 3.2 (3.5) | 3.8 (4.1) |
| | Asymmetry (%) | 0.96 [0.92–0.98] | 0.8 | 1.9 | 2.2 |
| Active range of motion: extension (°) | Dominant | 0.96 [0.92–0.98] | 1.2 (1.9) | 2.8 (4.3) | 3.4 (5.2) |
| | Non-dominant | 0.94 [0.88–0.97] | 1.4 (2.1) | 3.3 (4.8) | 4 (5.8) |
| | Asymmetry (%) | 0.89 [0.78–0.95] | 2.0 | 4.6 | 5.5 |
| Maximal isometric strength (N) | Dominant | 0.97 [0.94–0.99] | 7.4 (5.6) | 17.3 (13.0) | 20.6 (15.5) |
| | Non-dominant | 0.97 [0.93–0.99] | 7.1 (6.1) | 16.4 (13.9) | 19.6 (16.7) |
| | Asymmetry (%) | 0.73 [0.44–0.87] | 4.9 | 11.3 | 13.5 |

**Note:**
$N$ = 30. AV, absolute values expressed in the units of measurement; RV, relative values expressed as a percentage of absolute value; IC, confidence interval; ICC, Intraclass Correlation Coefficient; SEM, Standard Error of Measurement; MDC, Minimum Detectable Change; ECRB, extensor carpi radialis brevis; PPT, Pressure Pain Threshold.

(ICC: 0.79–0.82). For the pressure pain thresholds, reliability was 'excellent' for the extensor carpi radialis brevis point (ICC: 0.90–0.93), and 'good' for the deltoideus point (ICC: 0.84–0.89). For the active range of motion, reliability was 'excellent' for both extension, flexion, and total range (ICC: 0.94–0.97). Similarly, reliability was 'excellent' for the maximal isometric strength on both sides (ICC: 0.97). In the case of the asymmetries, reliability was 'excellent' for the active range of motion in flexion (ICC: 0.96); 'good' for the myotonometry parameters (ICC: 0.72–0.78), the Nine-Hole Peg test scoring (ICC: 0.72), the total and extension active range of motion (ICC: 0.73–0.89), and the maximal isometric strength (ICC: 0.73); and 'moderate' for the pressure pain thresholds on both points (ICC: 0.53–0.58).

## Interrelationship between tests

Correlation coefficients for muscle mechanical properties, manual dexterity, pressure pain thresholds, active range of motion, and manual isometric strength variables on both assessments for both sides and asymmetries are presented in the Supplemental Material. Consistently, 'strong' to 'moderate' positive correlations (*i.e.*, significant correlations on both assessments) were found for both the dominant and non-dominant sides between the myotonometry parameters of frequency and stiffness ($\rho > 0.875$; $P < 0.001$), and between pressure pain thresholds at the deltoideus and extensor carpi radialis brevis muscle points ($\rho > 0.520$; $P < 0.003$), and between the total active range of motion and active extension ($\rho > 0.674$; $P < 0.001$) and flexion ($\rho > 0.770$; $P < 0.001$) range of motion. Additionally, only for the dominant side, consistent 'moderate' positive correlations were found between the myotonometry parameters of decrement and the active flexion range of motion ($\rho > 0.518$; $P < 0.003$), and 'moderate' negative correlations were found between maximal isometric strength and the active total ($\rho < -0.612$; $P < 0.001$) and flexion ($\rho < -0.585$; $P < 0.001$) range of motion. Furthermore, consistent 'strong' to 'moderate' positive correlations

between the asymmetries for the myotonometry parameters of frequency and stiffness ($\rho > 0.919$; $P < 0.001$), and the total active range of motion and active extension ($\rho > 0.633$; $P < 0.001$) and flexion ($\rho > 0.491$; $P < 0.006$) range of motion were found. No significant correlations were found between the other variables after applying the correction for multiple comparisons.

## DISCUSSION

This study investigated the reliability of easy-to-implement upper limb assessments for muscle mechanical properties, manual dexterity, pressure pain thresholds, active range of motion, and maximal isometric strength, considering both absolute values and asymmetries between sides, as well as correlations between tests to determine their interrelationship. The results indicated 'good' to 'excellent' reliability for absolute values and 'moderate' to 'good' reliability for asymmetries. Although some intra-test correlations were observed, no inter-tests 'strong' correlations were found consistently. These findings indicate that a multimodal sensorimotor assessment test battery could provide rehabilitation professionals with valuable information to guide their monitoring of subjects with MSD in the upper extremities.

Previous research has investigated intra-limb reliability in healthy adults across various parameters of the forearm and hand. Concerning myotonometry variables (*i.e.*, oscillation frequency, dynamic stiffness, and logarithmic decrement), consistent 'excellent' reliability values were observed for forearm muscles, and specifically for the extensor carpi radialis brevis muscle (*Çevik Saldıran, Kara & Kutlutürk Yıkılmaz, 2022*). Similar reliability was also observed for the completion time of the Nine Hole Peg test, which has previously demonstrated 'good' to 'moderate' reliability and presenting faster performance on the second day compared to the first day (*Temporiti et al., 2022*), probably due to a learning effect. Moreover, the dominant hand performed faster than the non-dominant hand (*Bachman et al., 2023*). Therefore, to employ the Nine Hole Peg test effectively, it may be advisable for subjects to undergo a habituation session, or alternatively, consider discarding the first trials altogether.

Equally, pressure pain thresholds in healthy adults exhibited similar 'good' to 'excellent' reliability as in previous studies (*Pedersini et al., 2020*). With regards to active range of motion, previous studies showed 'excellent' reliability when assessing wrist extension and flexion (*Hanks & Myers, 2023*) as was found in this study. However, this study revealed higher ICC values for maximal isometric strength of the wrist extensors compared to previous studies (*Romero-Franco et al., 2019*). This discrepancy could be attributed to the stabilization of test performance using a strap, a unique approach in this study that has been shown to enhance measurement repeatability (*Custódio et al., 2023*).

Numerous studies have evaluated asymmetries as an outcome measure (*Baldursdottir et al., 2020*; *Gutiérrez-Espinoza et al., 2022*; *Jiménez-Del-Barrio et al., 2022*; *Burdukiewicz et al., 2020*; *Pradas et al., 2022*; *Bravo-Sánchez et al., 2019*), although the calculation methods employed sometimes deviate from current recommendations (*Parkinson et al., 2021*), potentially introducing biases (*Bailey, Sato & McInnis, 2021*). However, only few

studies have explored the reliability of asymmetry assessments (*Howe et al., 2020*; *Pérez-Castilla et al., 2021*), but none have done so specifically for the hand, forearm, or upper limb. When assessing changes over time, interventional or prospective studies frequently rely solely on interpreting *p*-values (*Andrade, 2019*), overlooking the necessity of contextualizing the magnitude of these changes by comparing them to reference values such as SEM or MDC (*Furlan & Sterr, 2018*). Determining whether the progression over time exceeds the measurement error values would aid in evaluating the clinical relevance of the recovery process and treatments used.

While certain variables, like the active range of motion, possess raw or absolute reference values approximating normality (*Soucie et al., 2011*), these values may not apply to individuals with physical-anatomical variations. For instance, for strength assessment, normalizing values by weight or body mass index (*McGee et al., 2020*) could complicate its interpretation and practical application. In this context, a notable advantage of employing asymmetry assessments is the practical relevance in offering clinically significant insights into a patient's or worker's functional status, without requiring a previous reference measurement for comparison. However, in athletes, where reference and follow-up measurements are common throughout the season, asymmetries can be considered adaptive in unilateral sports (*Ellenbecker et al., 2002*). Therefore, asymmetries should not be assessed in isolation but considered within a multidimensional context and considering the characteristics of the study population. For example, this study demonstrates that in a healthy population, the dominant side tends to outperform the non-dominant side in terms of strength and speed during fine motor skills.

The lower ICC values obtained for asymmetry assessments compared to raw values for both the non-dominant and dominant sides were to be expected as the asymmetry index includes the variance across time of two measurements instead of one. On the contrary, considering their acceptable reliability, asymmetry values should be interpreted with a conservative clinical perspective. In other words, this indicates that more substantial variations would be required to attribute the observed changes to factors other than measurement error, potentially indicating genuine improvement over time. In addition, it is worth noting the absence of systematic error between sessions for the values of the asymmetries. Consequently, rehabilitation professionals are advised not to limit their assessments and recordings to the affected side alone but to conduct a bilateral assessment, thereby obtaining a more comprehensive understanding of their clients' condition and progression.

The present study revealed 'strong' to 'moderate' positive intra-variable correlations like pressure pain thresholds, myotonometry parameters of frequency and stiffness, as well as the total active range of motion, active extension, and flexion range of motion. Nonetheless, consistent inter-variable correlations were largely absent among the multimodal variables, indicating each variable offers independent information, and potentially advocating for their inclusion in a comprehensive assessment of upper limb MSDs. An exception to this pattern was observed anecdotally on the dominant side, where 'moderate' positive correlations consistently emerged between myotonometry parameters

of decrement and the active flexion range of motion. This hints at a potential relationship between increased elasticity and enhanced flexibility, suggesting that greater elasticity may contribute to improved flexibility and *vice versa* (*Miyamoto & Hirata, 2019*). Interestingly, 'moderate' negative correlations were identified between the maximal isometric strength of wrist extensors and the active total and flexion range of motion. This finding indicates that individuals with higher strength in the wrist extensors tend to exhibit reduced active range of motion in flexion, possibly due to heightened passive resistance from antagonist muscles (*Nagano, Uoya & Nagano, 2019*). Overall, the intra-variable correlations and the lack of inter-variable correlations advocate for a streamlined sensorimotor assessment protocol, potentially involving a single parameter from each variable (*e.g.*, pressure pain threshold, maximal isometric strength, active range of motion and manual dexterity). Such an approach could prove sufficient to evaluate the clinical progression of the sensorimotor profile over time while minimizing assessment duration.

The main strength of this study is the implementation of a systematic assessment protocol to minimize the influence of confounding factors, controlling room conditions, and maintaining consistency in the timing of assessments. Additionally, providing MDC values at both 95% and 90% for both raw values and asymmetries offers greater utility to researchers and clinicians for monitoring improvements and calculating sample sizes. However, a primary limitation of this study is the utilization of data from a healthy volunteer cohort. Nonetheless, it can still establish reference values for assessing a normal range of asymmetries in clinical research or practice. Future research should apply this test battery to patient populations with MSD in the upper extremity to assess its responsiveness, accounting also for contextual and psychosocial factors, to evaluate its clinical feasibility.

## CONCLUSIONS

This study demonstrated 'good' to 'excellent' reliability in a test battery assessing muscle mechanical properties, manual dexterity, pressure pain thresholds, active range of motion, and maximal isometric strength of the hand and forearm from the dominant and non-dominant limb. However, when evaluating asymmetry values across limbs, the reliability ranged from 'moderate' to 'good'. These findings suggest that monitoring changes in asymmetries could offer a more conservative approach to detecting clinically substantial changes in individuals with hand or forearm injuries. Furthermore, certain variables showed consistent correlations within the same test, indicating potential interrelationships among them but not between different tests. These findings suggest that a multimodal sensorimotor assessment might be sufficiently comprehensive and efficient by considering only one test for each dimension (*i.e.*, muscle mechanical properties, manual dexterity, pressure pain thresholds, active range of motion, and maximal isometric strength of the upper limbs). Furthermore, to implement a biopsychosocial approach, it is recommended to combine the investigated tests with the monitoring of other variables such as the subject's subjective perception of recovery, the assessment of disability levels and psychological factors, or even the use of specific orthopedic examinations or imaging tests.

### Funding

Pablo Bellosta-López, Julia Blasco-Abadía, and Víctor Doménech-García are supported by Departamento de Ciencia, Universidad y Sociedad del Conocimiento, from the Gobierno de Aragón (Spain) (Research Group MOTUS B60_23D). Julia Blasco-Abadía is a predoctoral research fellow supported by the Grant PIF 2022-2026 from the Gobierno de Aragón (Spain). The funders had no role in study design, data collection and analysis, decision to publish, or preparation of the manuscript.

### Grant Disclosures

The following grant information was disclosed by the authors:
Departamento de Ciencia, Universidad y Sociedad del Conocimiento, from the Gobierno de Aragón (Spain): MOTUS B60_23D.
Gobierno de Aragón (Spain): PIF 2022-2026.

### Competing Interests

The authors declare that they have no competing interests.

### Author Contributions

- Pablo Bellosta-López conceived and designed the experiments, performed the experiments, analyzed the data, prepared figures and/or tables, authored or reviewed drafts of the article, funding adquisition, and approved the final draft.
- Julia Blasco-Abadía performed the experiments, analyzed the data, prepared figures and/or tables, authored or reviewed drafts of the article, and approved the final draft.
- Lars L. Andersen analyzed the data, authored or reviewed drafts of the article, and approved the final draft.
- Jonas Vinstrup analyzed the data, authored or reviewed drafts of the article, and approved the final draft.
- Sebastian V. Skovlund analyzed the data, authored or reviewed drafts of the article, and approved the final draft.
- Víctor Doménech-García conceived and designed the experiments, performed the experiments, analyzed the data, authored or reviewed drafts of the article, funding adquisition, and approved the final draft.

### Human Ethics

The following information was supplied relating to ethical approvals (*i.e.*, approving body and any reference numbers):

Comité de Ética de la Investigación de la Comunidad Autónoma de Aragón (C.P.-C.I. PI18/385).

### Data Availability

The raw measurements are available in the Supplemental Files. The raw data cover all variables presented in the manuscript.

## Supplemental Information

Supplemental information for this article can be found online at http://dx.doi.org/10.7717/peerj.17403#supplemental-information.

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
