# Peer review of "Multimodal sensorimotor assessment of hand and forearm asymmetries: a reliability and correlational study"

_PeerJ, doi:10.7717/peerj.17403_

## Round 0.1 · original submission · Minor Revisions

I follow both reviewers' specific suggestions and detailed comments, which should be considered to further improve the manuscript, especially with respect to clarity and consistency.

Reviewer 1 ·

Basic reporting

Overall, the text is clear and easy to read. However, there are minor inconsistencies or ambiguous language that could be improved:
1) the title can be expanded to give a detailed vision of the work done, also indicating the secondary objective, i.e., "A reliability and correlational study".
2) consistency between line 50, "Musculoskeletal disorders (MSDs) in the upper extremities", and lines 54-55, "upper extremity pain disorders".
3) it is not clear what it means in lines 60-61, "normal sensory function perceived by the individual.", or if the authors refer to "pain sensitivity" as in line 63.
4) In line 81, remove "methodological".
5) it is redundant to say, as in lines 85-86, "to determine the divergent validity or interrelationship between tests.". I suggest the authors maintain the term "interrelationship between tests" and remove the term divergent validity throughout the manuscript, as the last is more appropriate for assessing the psychometric properties of a questionnaire than a sensorimotor test.
6) consistency between terms to mention the personnel conducting the tests among days, i.e., sometimes assessor, evaluator, tester.
7) consistency between line 331, "multivariable sensorimotor assessment test battery", and the title and the rest of the text, "multimodal".

In Table 1, it is unclear what values were expressed as median [interquartile range: 25th-75th percentiles].

In order to make Table 2 self-explanatory, it would be helpful for the reader to add some symbols indicating the presence of a side effect or a day effect after the RM-ANOVA.

Experimental design

The description of all the procedures followed is exceptionally detailed and demonstrates a high degree of meticulousness in their execution.
However, it would be enriching to add to Figure 1 a third image with an extra angle to understand better the positionings (zenith, frontal, and lateral). Now, Figures 1A and 1B have a zenith perspective, but Figure 1A shows a lateral perspective, while Figure 1B shows a frontal perspective.
Consider also adding a picture exemplifying the actual procedure of each test as supplementary material. These images will be helpful not only to show the participant's position but also to show the assessor's positioning and execution of each test.

Validity of the findings

The presentation of the results and the discussion of the main issues are precise and detailed. Nevertheless:
1) the conclusion could benefit from more scope within the biopsychosocial model. It is stated in lines 429-430 that " which could be combined with the monitoring of the subject's subjective perception of recovery.". It would seem more appropriate also to name, depending on the context, other complementary variables, such as the assessment of disability levels, the assessment of psychological factors, or even the use of specific orthopedic tests or imaging tests, such as ultrasound, to have a complete record of the participant's, patient's, worker's or athlete's condition.
2) the discussion could benefit from a few sentences highlighting the absence of systematic error between sessions for the values of asymmetries, as presented in the results lines 283-284.

Additional comments

Although the report focuses on several aspects of the manuscript that need to be improved, there are many other elements to highlight and acknowledge. Your meticulous and rigorous research can be useful for staff in various fields. Congratulations on the comprehensive and detailed work you have conducted.

Reviewer 2 ·

Basic reporting

Thank you for the opportunity to review your work. Overall, this is a well written manuscript.

Abstract
Line 30. Sentence end in a preposition. Edit to read “….between sessions.”
Lines 35-44, results. In the interest of brevity, perhaps the authors should consider removing the ICC, SEM, and MDC data since it is well documented in the manuscript and table 3. Likewise, the description of how the asymmetries were calculated does not need to be in the abstract.

Manuscript introduction
Well done

Experimental design

Materials and Methods
Line 117. Delete “Beside”
Line 132. Abstract mentions the retesting in 1 week but there is no mention of it in the manuscript procedures. Were the participants advised to control their activity between testing sessions?
Line 137, testing positions. The use of the term shoulder level is not consistent with the figure. “Shoulder level” implies the humerus is at the level of the shoulder, i.e. parallel to the table. Based on the picture, the shoulders are in slight flexion.
Line 169. Company location for Smith & Nephew

Validity of the findings

Results
Well done

Discussion
Lines 348-351. This does not make sense to me. Could you entertain saying this another way?
Line 375. Is this “or” or “and”?
Line 379. “our” appears to be a typo. It does not make sense.
Line 401. You mention streamlining the assessment, might you provide an example?
Line 413-415. Assessing reliability using the current protocol with a patient population could prove problematic since status of people with pathology is likely to change over the course of a week. Reliability may need to be done on the same day.

Additional comments

Tables - Figures
Table 1. What was the point of assessing the arthropometric characteristics? The data was reported but never used to relate to the assessment techniques. Did you have a theory about these relationships?
Table 2. Very good
Table 3. Very good
Figure 1. Very helpful

---

## Round 0.2 · accepted · Accept

The authors have satisfactorily addressed all comments raised by both reviewers. Although the initial reviewers were not invited to evaluate this revised version due to solely minor revisions, I have checked the modifications made in response to their feedback and can confirm that all points mentioned have been sufficiently addressed.